# Nanoindentation and Microstructure in the Shear Band in a Near Beta Titanium Alloy Ti-5Al-5Mo-5V-1Cr-1Fe

**DOI:** 10.3390/ma12244065

**Published:** 2019-12-05

**Authors:** Bingfeng Wang, Xu Ding, Ying Mao, Lanyi Liu, Xiaoyong Zhang

**Affiliations:** 1State Key Laboratory for Powder Metallurgy, Central South University, Changsha 410083, China; wangbingfeng@csu.edu.cn (B.W.); 153112135@csu.edu.cn (L.L.); 2School of Materials Science and Engineering, Central South University, Changsha 410083, China; dingxu-mse@csu.edu.cn (X.D.); maoyingmao97@csu.edu.cn (Y.M.)

**Keywords:** titanium alloy, shear band, nanoindentation, deformation, microstructure

## Abstract

Shear localization is the main deformation mode for the near beta titanium alloy Ti-5Al-5Mo-5V-1Cr-1Fe loaded at high strain rates at either room temperature or cryogenic temperature. Nanoindentation, transmission electron microscopy, and high-resolution electron microscopy technique are applied to character the microstructure features and mechanical properties in the shear band of near beta titanium alloy. A white and straight band is observed in the shear region. Both microhardness and nanoindentaion hardness in the shear region are inferior to those in matrix. The different microstructure in the edge and the center in the shear band contribute to different mechanical properties. The plasticity of the entire shear band is almost homogenous when specimens are deformed at the cryogenic temperature. Rotational dynamic recrystallization is responsible for the formation of the ultrafine grains in the shear band. The edge of the shear band is composed of elongated grains, while there are ultrafine equiaxed grains in the center of the shear band. Deformation temperature has significant influence on the process of the grain refinement and the phase transformation in the shear band (SB). The grain sizes of the shear band in the specimen deformed at room temperature are larger than those in the specimens deformed at cryogenic temperature. The shear band consists of α phase grains in the specimen deformed at room temperature, and the shear band consists of α phase and lath-like α′ phase grains in the specimen deformed at cryogenic temperature. Finally, the mechanisms for phase transformation in the shear band are illustrated.

## 1. Introduction

Ti-5Al-5Mo-5V-1Cr-1Fe (Ti-55511) is a typical near beta titanium alloy with attractive combinations of mechanical properties such as super intensity to weight ratios and toughness, which is often applied in the aerospace industry and engineering structural applications [1,2]. There are also many unique properties such as super plasticity and high strength when it is processed and employed at cryogenic temperature. Furthermore, Ti-55511 has gradually become a promising material in the cryogenic temperature engineering field [3,4,5].

A narrow zone called adiabatic shear band (SB) generates when the specimens were deformed under dynamic loading conditions such as laser shock, high velocity cutting, and explosive fragmentation [6]. Ultrafine grains and nanograins in the center of the SB were observed [7]. Mechanism for microstructure formation of the SB attracted lots of scientists. For research methods, the extensive observations of the microstructure were conducted by using an optical microscopy (OM), electron back scattering diffraction (EBSD), transmission electron microscopy (TEM), and nanoindentation. Dynamic recrystallization mechanism was responsible for the generation of ultrafine equiaxed grains in the SB in pure titanium [8]. Dynamic recovery contributed to the formation of ultrafine grain with heavy dislocations in the SB of the coarse grain Ti-55511 alloy [9]. Phase transformation was also a typical characteristic in the SB of high alloying near beta titanium alloy Ti-1300 at dynamic loading conditions [10]. It was a consensus that rotational dynamic recrystallization contributed to the kinetics possibility in the generation of ultrafine equiaxed grains in the SB, especially for titanium [11] and titanium alloy [12,13,14]. 

The previous research [15] mainly focused on the width of the SB and the grain refinement induced by rotational dynamic recrystallization mechanism in the SB in the Ti-55511 titanium alloy. However, the difference of the mechanical properties in the center and in the edge of the SB in the Ti-55511 titanium alloy is unclear. Moreover, the phase transformation characteristics and the microstructure characteristics influenced by deformation environment temperature in the Ti-55511 titanium alloy are not well concluded, especially at a cryogenic temperature. 

The objectives of our research are to investigate the mechanical properties of the SB by the microhardness and the nanoindentation, and to characterize the microstructure features in the SB, especially the phase transformation, and to discuss the influence of the deformation temperature on the formation of the microstructure in the SB in the near beta titanium alloy Ti-55511.

## 2. Materials and Experiments

The chemical compositions (wt.%) of the as-received Ti-55511 titanium alloy are shown in Table 1. The Ti-55511 titanium alloy with an initial microstructure of fine α phase grains with sizes of about 5 μm distributed within β phase matrix with grain sizes of about 20 μm as shown in Figure 1 was prepared for the current study. The grain size was measured by the software Image Pro Plus (Version 6). The dynamic loading tests were carried out in a split Hopkinson pressure bar (SHPB) system (Beijing Xiaotu Technology Co., Ltd, Beijing, China) with a thermostat box to keep the experimental environment at cryogenic temperature (173 K) or room temperature (293 K), as shown in Figure 2a. We prepared hat-shaped specimens to obtain well-controlled SBs. The schematic image in Figure 2b shows the size of the hat-shaped specimen. Figure 2c shows a picture of deformed hat-shaped specimens after dynamic tests under cryogenic temperature and room temperature, which are labeled as CT specimen and RT specimen, respectively. All of the specimens were impacted along the impact direction in Figure 2a and each of them was cut into two halves along the main axis after dynamic tests.

The selected surfaces of the SB were polished and etched by the Kroll reagent (1 mL hydrofluoric acid + 3mL nitric acid + 50 mL water). OM was conducted by a POLYVAR-MET optical microscope (Leica, Solms, Germany). For reliable hardness data, the selected surfaces were mechanically ground and subsequently polished with diamond paste. Microhardness was tested by HMV-2T (SHIMADZU, Kyoto, Japan) with a pressure of 100 g and a holding time of 10 s. Nanoindentaion tests were completed by Nano Test Vantage (Micromaterials, London, UK), equipped with a Berkovich indenter (Micromaterials, London, UK). Nanoindentation tests with a maximum force of 50 mN and a loading rate of 1 mN·s^−1^ were applied to the SB and the α phase and β phase grains in the matrix. To get mechanical properties of ultrafine grains in the SB, nanoindentation tests with a maximum force of 5 mN and a loading rate of 3 nm·s^−1^ were processed in the edge and center of the SB. The indenter was held for 300 s during the holding stage to determine creep behaviors. Both microhardness and nanoindentation tests were conducted at room temperature. After nanoindentaion tests, the backscattered electron pictures around the indents were obtained by FEI Quanta-200 (FEI, Amsterdam, Netherlands).

TEM samples were accurately cut from SBs by the method of a focused ion beam (FIB) instrument. First, specimens were cut out to foils with a thickness of 0.5 mm along the loading axis by line cutting. Then we mechanically polished it into a thickness of less than 80 μm. Finally, these specimens were cut into a side length of 10 μm square by FIB. The bright electron images were obtained by the Tecnai G2 20 ST transmission electron microscope (FEI, Amsterdam, Netherlands) carried out at 200 kV and the Titan G2 60-300 high resolution transmission electron microscope (FEI, Amsterdam, Netherlands) carried out at 300 kV, respectively.

## 3. Results

### 3.1. Mechanical Properties of the Shear Band

Optical micrographs in Figure 3 demonstrate that long and straight white-etching bands in the SB of the specimens deformed at cryogenic temperature and room temperature are generated. The SBs are clearly separate to the matrix by obvious boundaries, and severely elongated grains along the boundaries can be observed in both specimens. The width of the SB in the RT specimen is wider than that of the CT specimen.

Figure 4a,b shows the optical micrographs of microhardness in the SB and the matrix in the CT and RT specimen, respectively. Indentation positions on the SB and on the matrix in both specimens can be clearly observed. Results in Figure 4c,d show the matrix have higher microhardness than the SB in both Ti-55511 titanium alloy specimens. Ran et al. [16] also investigated the microhardness on the SBs in the Ti-55511 alloy and achieved similar results.

Figure 5a,b shows the backscattered electron images of the nanoindentation positions in the SB and the matrix in the CT specimen and the RT specimen, respectively. Indentation positions on the SB and the α phase grains and the β phase grains are marked by black dotted circles. Furthermore, the size of indentations is about 5 μm when specimens were deformed under the maximum force 50 mN. Our studies demonstrate that the nanoindentation hardness in the SB is lower than that in the matrix, and therefore finer, as shown in Figure 5c,d.

The grain sizes in the SB are often lower than those in the matrix [11]. The Hall–Patch relationship tells that smaller grain sizes result in higher strength and hardness. That is to say, the values of the microhardness and the nanoindentation hardness in the SB should be higher than those in the matrix. However, in the present research, after excluding the problems during the microhardness and the nanoindentation measurements, we find that the values of the microhardness and the nanoindentation hardness in the SB in both near β titanium alloy Ti-55511 specimens are lower than those in the matrix. During the shear localization, not only the grain refinement but also the phase transformation occurred [7]. The different microstructure in the SB and the matrix determine mechanical performance. Therefore, it is the phase compositions rather than grain sizes that influence the mechanical performances in the SB and the matrix.

Moreover, Figure 5c,d shows that the nanoindentation hardness of the α phase grains is much higher than that of the β phase grains in matrix. The crystallographic structure of β phase grain is body-centered cubic (BCC) crystal structure. However, α phase grain is hexagonal close-packed (HCP) crystal structure. It is known that the numbers of the available slip system in a BCC crystal structure are larger than those in a HCP crystal structure [17]. Therefore, the deformation is easier for the BCC crystal structure. It is reported that the nanoindentation hardness inside the grains was lower than that near the grain boundaries due to the grain boundaries strengthening effect [18]. Due to the different scales of the β phase grain sizes and nanoindentaion, it is unclear whether the deformation area is inside the β phase grains or in the β phase grain boundaries. The nanoindentation hardness of the β phase grains in the matrix shows a greater scatter than that of the α phase grains in the matrix.

Figure 6a,b shows the backscattered electron images of the nanoindentation positions in the SB of the CT specimen and the RT specimen, respectively. Indentation positions in the SB are marked by white dotted circles. Furthermore, sizes of indentations are about 1 μm when specimens were tested at the maximum force 5 mN. Figure 6c,d shows typical force (F)–displacement (h) curves in the edge and the center of the SB in the CT specimen and the RT specimen, respectively. Results show that the gaps of maximum depths between the edge and the center of the SB in the CT specimen are lower than those in the RT specimen. Figure 7a,b shows representative creep behavior curves in the edge and in the center of the SB during nanoindentation holding stage, respectively. Moreover, the fitted curves are plotted in the figures obtained as the reference [19]. The creep displacement increases with holding time until it is balanced. Creep velocity illustrates the plasticity of materials under creep deformation [20]. Both plasticity and hardness represent the mechanical properties related to the microstructure. There are balances between strength and plasticity in metals [21]. The creep velocity of the grains in the edge and the center of the SB in the CT specimen are nearly the same, as shown in Figure 7a. However, Figure 7b shows the creep velocity of grains in the center of SB is lower than that of grains in the edge of the SB in the RT specimen. The results show that the plasticity of the entire SB in the CT specimen is almost homogenous. Figure 7c shows the comparison of the nanoindentation hardness values for the SB and the matrix in the CT specimen and the RT specimen, respectively. It illustrates that the nanoindentation hardness values of the center are larger than those of the edge in the SB. The values of nanoindentation hardness for the center and the edge of the SB in the CT specimens are about 6.25 GPa. The values of nanoindentation hardness for the center and the edge of the SB in the RT specimen are about 7.5 GPa and 5.75 GPa, respectively. The different microstructure in the edge and the center of the SB contribute to the difference in mechanical properties of the specimens. Furthermore, the SBs possess a different microstructure because the specimens were deformed at cryogenic temperature or room temperature.

### 3.2. Microstructure in the Shear Band

The bright electron images in Figure 8 demonstrate the microstructure in the SB of the specimens after the specimens deformed at cryogenic temperature or room temperature. There are composed of elongated grains in the edge of the SB, and the center of the SB is made up of ultrafine grains in both CT specimen and RT specimen. Features of these ultrafine grains are polygonal and defined grain boundary, which are typical characteristics of dynamic recrystallized grains [22]. Grain sizes of the elongated grains along the boundaries of SB are about 300 nm and 500 nm in the CT specimen and the RT specimen, respectively. Grain sizes of the ultrafine grains in the center of the SB are about 100 nm and 200 nm in the CT specimen and the RT specimen, respectively. Therefore, the grain sizes in the SB of the RT specimen are larger than those in the SB of the CT specimen. It also can be found that there are amounts of lath-like microstructure in both the edge and the center in the SB of the CT specimen. However, this lath-like microstructure cannot be observed in the RT specimen.

Figure 9 shows crystallographic microstructure of the grains in the SB captured by high resolution electron microscopy (HREM). Figure 9a illustrates the microstructure in the SB of the CT specimen. Figure 9b,c shows high resolution electron images and corresponding fast Fourier transform (FFT) images (digital diffractograms) for the ultrafine grain and the lath-like grain. Figure 9b shows the enlarged area in Figure 9a marked in white solid lines, and Figure 9c shows the enlarged area in Figure 9a marked in white dotted lines. By calibrating the digital diffractograms, we can find both digital diffractograms are hexagonal crystal structure. The crystal structure of α′ phase and α phase in the titanium alloy are hexagonal crystal structure [23]. Furthermore, the typical characteristic of α′ phase in titanium alloy is a lath-like structure. Therefore, the SB in the CT specimen consists of α′ phase and α phase. Figure 9d shows the microstructure in the SB of the RT specimen. Figure 9e,f shows high resolution electron images and corresponding fast Fourier transform (FFT) images (digital diffractograms) for the ultrafine grain and the elongated grain. Figure 9e shows the enlarged area in Figure 9d marked in white solid lines, and Figure 9b shows the enlarged area in Figure 9a marked in white dotted lines. By calibrating the digital diffractograms, we can find both digital diffractograms are hexagonal close-packed crystal structure. It indicates that there are α phase grains in the SB of RT specimen. Due to the difference in the grain sizes, the nanoindentation hardness in the center is larger than that in the edge of the SB in the RT specimen. A mixture of the α′ phase grains and the α phase grains not only reduced the nanoindentation hardness values and the effect of the grain sizes, but also made the mechanical properties of the microstructure of the SB keep nearly homogenous levels.

## 4. Discussion

By analyzing the microstructure evolution and mechanical properties above, we can conclude that phase transformation happens in the SB and the surrounding deformation conditions influence the phase transformation process. In our previous paper [15], we expressed the true stress versus true strain curves for the present specimens. Both specimens have the same deformation history during dynamic loading. Temperature rising within adiabatic shear localization related to the deformation holds an important role in the microstructure mechanism research. Because of the extremely short time during high strain rate (>10^3^ s^−1^) loading, the shear localization process can be considered as an adiabatic process. The following equation [12] can be applied for the calculation of adiabatic temperature rising.
∆T = T − T_0_ = β/(ρ·C_v_)·∫σdε,(1)
where T_0_, C_V_, ρ, ε, and σ are the deformation environment temperature, the heat capacity, the constant of mass density, the true strain, and the true stress, respectively. β denotes the ratio of plastic energy converted to heat, and usually β is 0.9. Furthermore, T_0_ is 173K and 293K. After the consideration for the true stress versus true strain curves [15], it can be calculated that the maximum temperatures in the SB are about 1346 K and 1340 K for the CT specimens and the RT specimens, respectively. The recrystallization point of the Ti-55511 titanium alloy is treated as 0.4–0.6 T_m_ (720–1080 K) and phase transformation occurs at about 1150 K. Therefore, the temperature rising in the SB satisfies the conditions for the phase transformation and recrystallization of the Ti-55511 titanium alloy. 

Based on the rotational dynamic recrystallization mechanism, the generation of new grains needs local grain boundaries rotation driven by minimizing the interfacial energy [24]. Moreover, the local grain boundary segments can have a rotation of about 30° during the deformation process. The following equation can be used to calculate the process of the time needed [25].
t = L·k·T·f(θ)/(4δ·η·D_b0·_exp(−Q/RT)),(2)
where t, L, δ, and η stand for the time, the average subgrain diameters, grain boundary thickness, and the grain boundary energy, respectively. D_b0_ denotes a constant decided by grain boundary diffusion. Q_b_ represents the activation energy for grain boundary diffusion and Q_b_ equals to (0.4–0.6) Q and Q denotes the activation energy for grain growth. θ represents the subgrain misorientation angles. The following equation can be used to calculate f(θ) [26].

f(θ) = (3tanθ − 2cosθ)/(3 − 6sinθ) − (1 − ln((2 + √3)/(2 − √3)))·4√3/9 + ln((tan(θ/2) – 2 − √3)/(tan(θ/2) – 2 + √3)) + 2/3.(3)

For titanium alloys [12], the values of δ, η, D_b0_, Q, k, and R are 5.8 × 10^−10^ m, 1.19 J·m^−2^, 2.8 × 10^−5^ m^2^·s^−1^, 312 kJ·mol^−1^, 1.38 × 10^−23^ J·K^−1^, and 8.314 J·K·mol^−1^, respectively.

From Figure 8, the average subgrain diameters L in the SB are 100 nm and 200 nm for the CT specimen and the RT specimen. By substituting the parameters into Equations (2) and (3), the kinetic curve of the rotational dynamic recrystallization mechanism in the SB can be gained. Through the combination of the analysis of the temperature and the kinetic curve, we obtain the comprehensive curves for phase transformation in the CT specimen and the RT specimen, as shown in Figure 10a,b, respectively. It can be seen that either the CT specimen or the RT specimen can finish the generation of the ultrafine grains in several microseconds by the rotational dynamic recrystallization mechanism, and temperature in the SB is enough to begin phase transformation. For the near β titanium alloy Ti-55511, the common ways for phase transformation are β phase to α′ phase or α′′ phase and then to α phase. Due to the difference in the surrounding deformation temperature, the cooling rate for the specimen deformed at cryogenic temperature is a magnitude times larger than that for the specimen deformed at room temperature. Therefore, the phase transformation breaks in the SB of the CT specimen and the mixture microstructure of α phases and α′ phases is generated. 

The schematic diagrams of the process of the grain refinement induced by rotational dynamic recrystallization in the SB are illustrated in Figure 11. Four steps are used to demonstrate the formation of the ultrafine grains in the SB. The original grains are firstly elongated under the shear stress in Figure 11a. As it is shown in Figure 11b, large amounts of dislocations generate inside the elongated grains due to the severe deformation. Then the dislocations transfer into subgrain boundaries and subgrains form as it is presented in Figure 11c. Finally, the subgrain boundaries rotate about 30° and subgrains break into ultrafine grains in the SB in Figure 11d. 

## 5. Conclusions

Near beta titanium alloy Ti-55511 possesses strong sensitivity to shear localization and generates SBs under dynamic loading at either cryogenic temperature or room temperature. The SB is straight and white band. The microhardness and the nanoindentation hardness in the matrix are higher than those in the SB because of the different phase compositions in the matrix and the SB. The nanoindentation hardness of the α phase grain with HCP crystal structure is higher than that of the β phase grain with BCC crystal structure in the matrix. The creep velocity of the grains in the edge and the center of the SB in the CT specimen are nearly the same. However, for the RT specimen, the creep velocity of grains in the edge of the SB is larger than that in the center of the SB. The plasticity of the entire SB in the CT specimen is almost homogenous. The nanoindentation hardness values for the center and the edge are about 6.25 GPa in the SB of the CT specimen, respectively. The nanoindentation hardness values for the center and the edge are about 7.5 GPa and 5.75 GPa in the SB of the RT specimen, respectively. The nanoindentation hardness of the edge is lower than that of the center in the SB due to the finer grains in the center and the same phase compositions in both the edge and the center of the SB. 

The different microstructure in the edge and the center of the SB contribute to different mechanical properties. Rotational dynamic recrystallization is responsible for the grain refinement in the SB. The edge of the SB is composed of elongated grains, and the center of the SB consists of ultrafine equiaxed grains. Deformation temperature has significant influence on the process of the grain refinement and the phase transformation in the SB. Grain sizes of the SB in the specimen deformed at room temperature are averagely larger than those of the SB in the specimen deformed at cryogenic temperature. The SB in the CT specimen consists of lath-like α′ phase and α phase. The grains in the SB in the RT specimen are α phase grains. 

## Figures and Tables

**Figure 1 materials-12-04065-f001:**
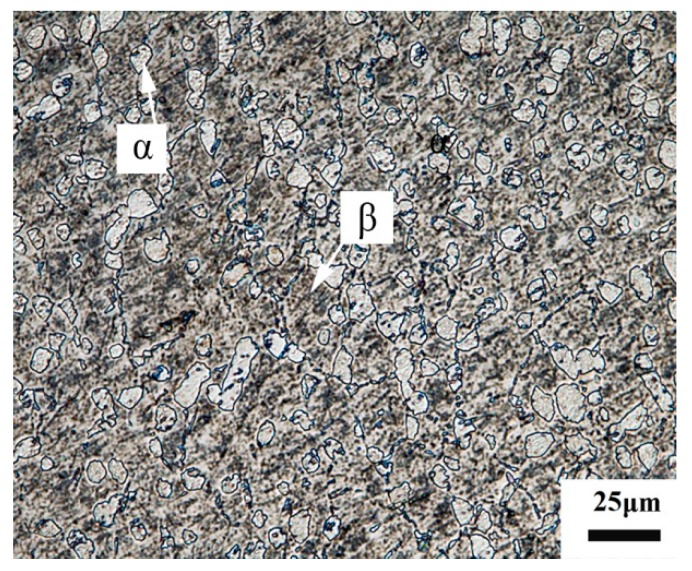
Initial optical micrograph of the Ti-55511 titanium alloy.

**Figure 2 materials-12-04065-f002:**
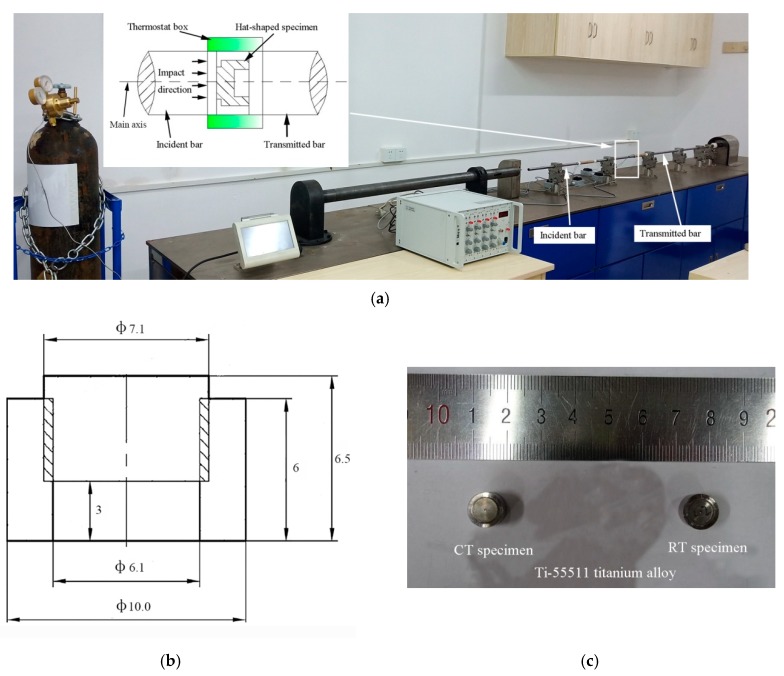
(**a**) Split Hopkinson pressure bar (SHPB) test system; (**b**) schematic image of the hat-shaped specimen; (**c**) pictures for cryogenic temperature (CT) specimen and room temperature (RT) specimen.

**Figure 3 materials-12-04065-f003:**
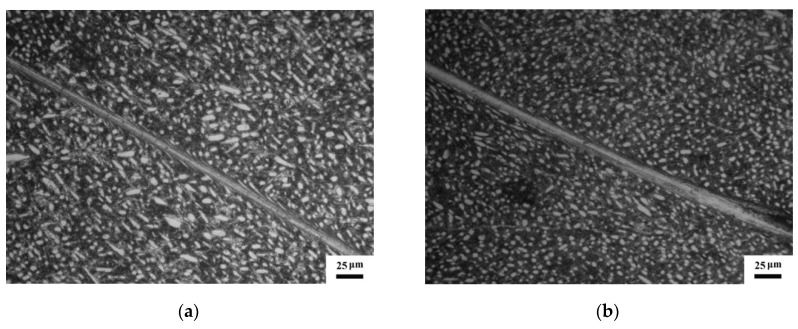
Optical micrographs of the shear regions: (**a**) CT specimen, (**b**) RT specimen.

**Figure 4 materials-12-04065-f004:**
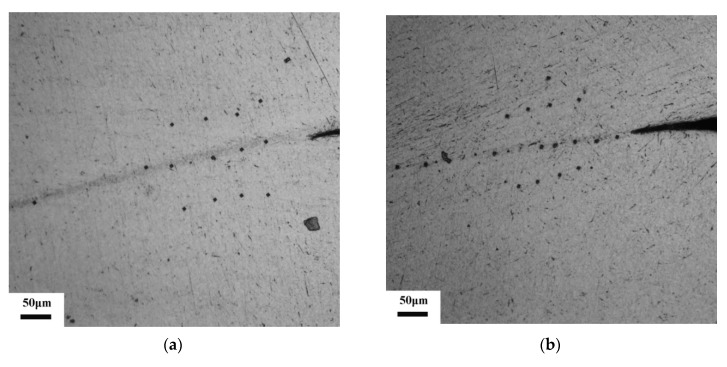
Optical micrographs of microhardness positions with a pressure of 100 g: (**a**) CT specimen; (**b**) RT specimen. The microhardness in the shear band (SB) and matrix: (**c**) CT specimen; (**d**) RT specimen.

**Figure 5 materials-12-04065-f005:**
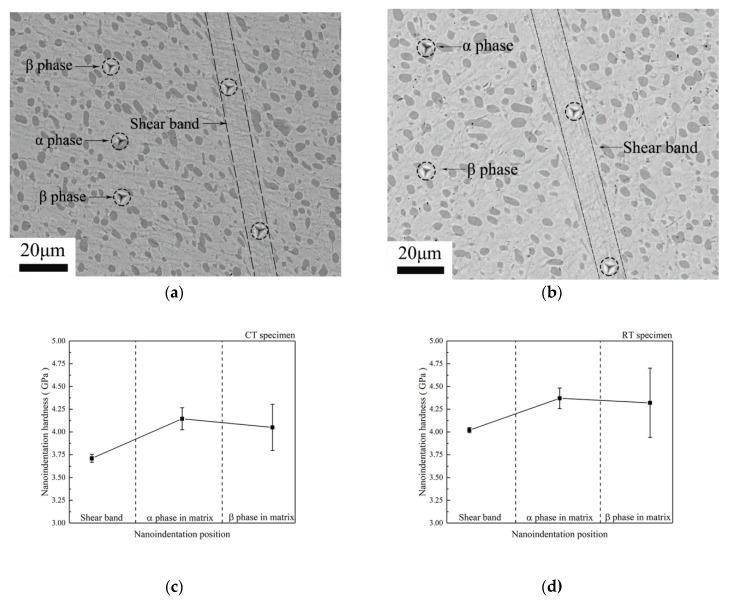
Backscattered electron images of nanoindentation position with a maximum force of 50 mN: (**a**) CT specimen; (**b**) RT specimen. The nanoindentation hardness in different position: (**c**) CT specimen; (**d**) RT specimen.

**Figure 6 materials-12-04065-f006:**
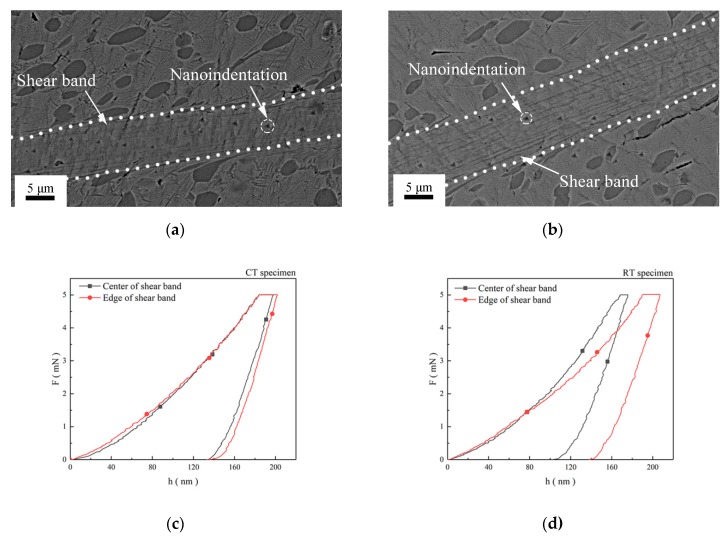
Backscattered electron images of nanoindentation position with maximum force of 5 mN: (**a**) CT specimen, (**b**) RT specimen. Nanoindentation force (F)–displacement (h) curves: (**c**) CT specimen, (**d**) RT specimen.

**Figure 7 materials-12-04065-f007:**
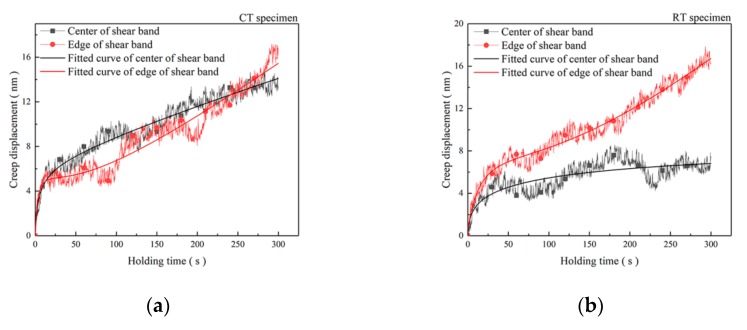
Creep displacement–holding time curves: (**a**) CT specimen, (**b**) RT specimen. (**c**) Nanoindentation hardness in the SB.

**Figure 8 materials-12-04065-f008:**
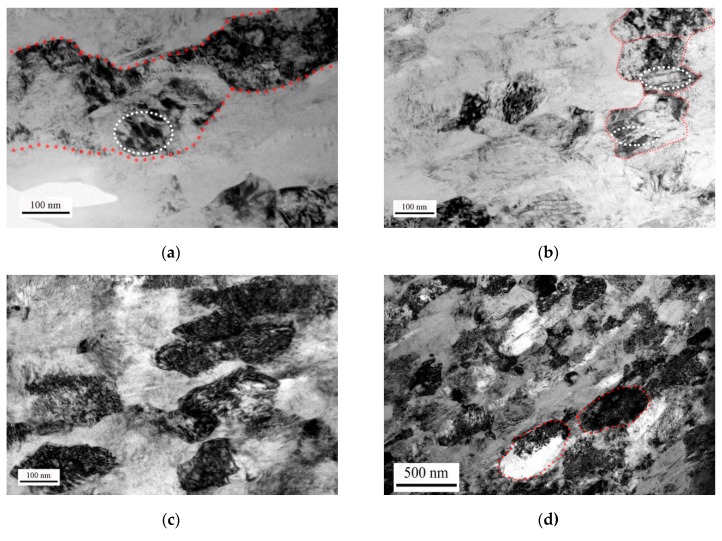
Bright electron images of microstructure in the SB. (**a**) TEM image along boundary of the SB in the CT specimen, (**b**) and (**c**) are TEM images in the center of the SB in the CT specimen. (**d**) TEM image along boundary of the SB in the RT specimen, (**e**) TEM image in the center of the SB in the RT specimen.

**Figure 9 materials-12-04065-f009:**
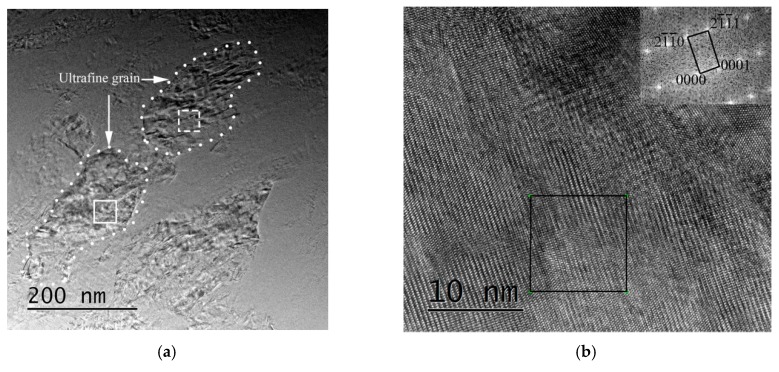
High resolution bright electron images of the microstructure in the SB. (**a**) High resolution electron microscopy (HREM) image in the SB in the CT specimen, (**b**) and (**c**) are the HREM images and corresponding selected area diffraction (SAD) pattern insets after FFT (Fast Fourier Transformation) in the ultrafine grain and the lath-like grain, respectively. (**d**) HREM image in SB in the RT specimen, (**e**) and (**f**) are the HREM images and corresponding SAD pattern insets after FFT in the ultrafine grain and the elongated grain, respectively.

**Figure 10 materials-12-04065-f010:**
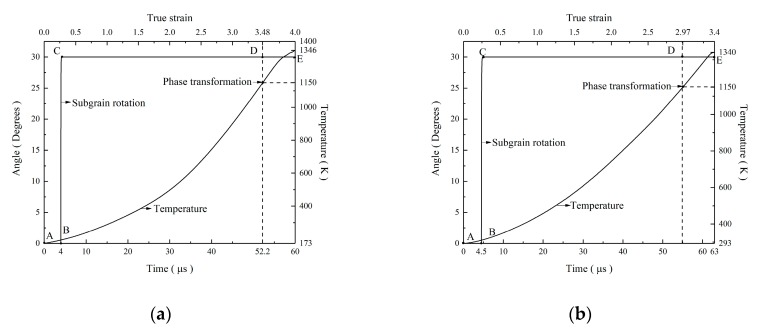
(**a**) Angle of subgrain rotation and temperature as a function of true strain and time when subgrain sizes L = 100 nm in the SB of the CT specimen. (**b**) Angle of subgrain rotation and temperature as a function of true strain and time and true strain when subgrain sizes L = 200 nm in the SB of the RT specimen.

**Figure 11 materials-12-04065-f011:**
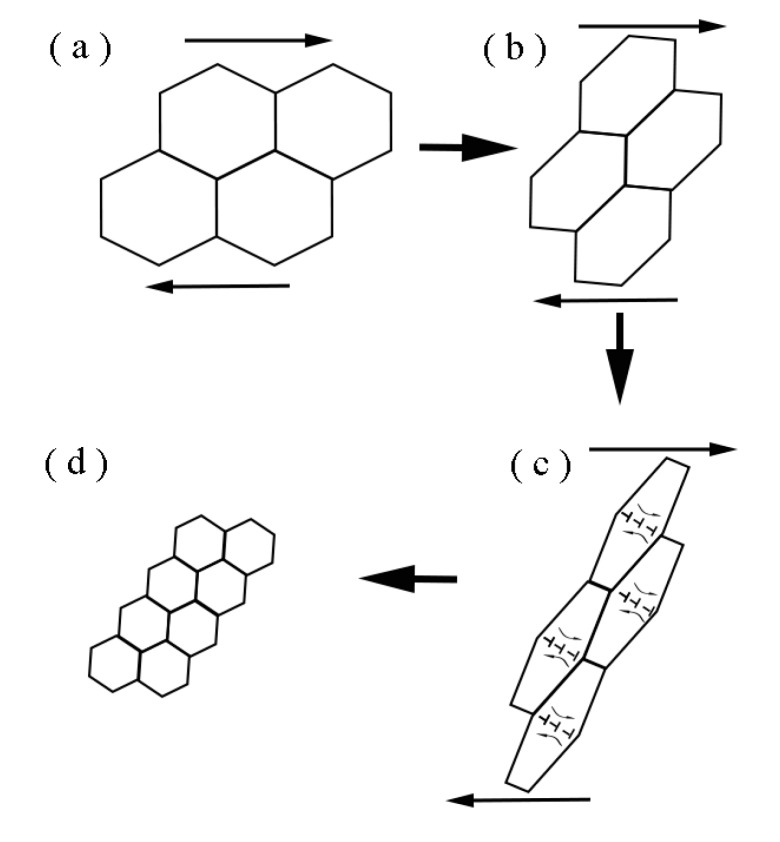
The schematic diagram of the process of the grain refinement in the SB.

**Table 1 materials-12-04065-t001:** The chemical compositions (wt.%) of the as-received Ti-55511 titanium alloy.

Elements	Al	Mo	V	Cr	Fe	Ti
Compositions (wt.%)	5.59	5.31	4.24	1.27	1.18	Balance

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
