# Peer review of "Nanoindentation and Microstructure in the Shear Band in a Near Beta Titanium Alloy Ti-5Al-5Mo-5V-1Cr-1Fe"

_materials, 2019, doi:10.3390/ma12244065_

Round 1

Reviewer 1 Report

The article describes microstructure and nanoindentation properties of the SB detected in the near beta Ti alloy. The results presented are enough scientifically interesting to be publish in the Journal, however after revision. Detailed remarks are presented below.

In my opinion the title should be changed: Nanoindentation and microstructure observations of the shear band in a near beta titanium alloy Ti-5Al-5Mo-5V-1Cr-1Fe.

Line 59. Show a general photo of the split Hopkinson pressure test stand used for investigation.

Line 65. Fig. 1 is not acceptable, there is no grain size visible. How did you measure the grain size of the b phase?

Line 67. Fig. 2. Draw the main axis in Fig. 2a, show the size of the hat-shaped specimen tested and enlarge the image of the specimens presented in Fig. 2b.

Line. 71. What was the temperature of the nanoindentation testing?

Line 97. There is no explanation (comments) about the higher results of the microhardness data measured for the RT sample (~375 HV) in comparison with the CT one (~325 HV). The same question is related to the nanoindentation results presented in Fig. 5c and 5d.

Line 104. ”The results of microhardness and nanoindentation hardness show different regularity with Hall-Patch relationship, which tell smaller grain size result in higher strength and hardness” This sentence is very general and should be discussed more profound, including problems generated during nanoindentation measurements.

Line 109. “Also, Figures 5c,d show that the nanoindentation hardness of the α phase grains are much higher than those of the β phase grains in matrix”. One can observed a greater scatter of the nanoindentation hardness of the β phase matrix, in comparison with the α phase, but there is no explanation about that.

Line 111. “The reason of different nanoindentation hardness for the α phase and the β phase grains in matrix is the difference in crystallographic structure”. This issue should be discuss more detailed.

Line 128. There is no explanation why the creep velocity of grains in the edge of the SB is larger than that in the center of the SB for the RT specimen?

Line 174. Fig. 9. It should be marked, where from the HRTEM images were taken?

Author Response

The responses to the review comments are in the attachment file.

Reviewer 2 Report

This paper presents results on the relationship between mechanical properties and phase transformation in near beta Ti-55511 alloy. The aim of the paper is interesting and suitable in my opinion to Journal scope. The article is clear and well written, although, in the present form, some improvements are required:

The introduction need to be extended. Furthermore, add references on this context. Add details of novelty of this article compared to literature reference. In particular a clear distinction with “Materials Science and Engineering: A, 736, 24, 202-208 (2018)”. Focus the attention on the improvement of knowledge induced by this experimental investigation, proposed in this paper, compared with the result and discussion reported in the previous article. In “Materials and Experimentals” add chemical composition of the alloy. Please modify figure 4c and figure 4d in order to have a same y axis (i.e. max at 375 HV) to better evidence the differences. Apply a similar approach on figure 5. A schematic diagram of the process of the grain refinement induced by shear bands can improve the quality and readability of the paper. The discussion need to be improved better clarifying the micro-structural differences between RT and CT clarifying as these influence the mechanical performances.

Author Response

The responses to the review comments are in the attached file.

Round 2

Reviewer 1 Report

The article still has some typewriting and English grammar mistakes and remarks that have to be corrected. Some of them are listed below.

Line 65-67. Fig. 2c is not described in the text.

Line 107. “Ran et al. [16] also investigated the microhardness on the SBs in the Ti-55511 alloy and had the same results with us achieved similar results”.

Line. 109. Enlarge graphs in Fig. 4c, 4d, 5c and 5d. They are blurred.

Line 125. Correct the word influent for influence: …it is the phase compositions rather than grain sizes that influent influence the mechanical performances in the SB and the matrix.

Line 127. ” Also, Figures 5c,d show that the nanoindentation hardness of the α phase grains are much higher than those of the β phase grains in matrix. The reason of different nanoindentation hardness for the α phase and the β phase grains in matrix is that β phase grains are softer than α phase grains in titanium alloy.” Read this text carefully. The second sentence is obvious and redundant. Delete it.

Line 257. Correct English grammar “Large amounts of dislocations are generated inside the elongated grains due to the severe deformation accumulated (Fig. 11b)”.

Line 259. Then the dislocations transfer into subgrain boundaries and subgrains form as it is presented in Figure 11c.

Line 272. “…the nanoindentation hardness in the matrix are is higher …”.

Line 277. Conclusions should be supplemented with new information improved in the text, e.g. explanation about the difference in the nanoindentation hardness.

Author Response

Dear Reviewer:

  Please find the responses in the attached PDF.

Thank you!

Best regards.

Authors
